# Fatal Infection in an Alpaca (*Vicugna pacos*) Caused by Pathogenic *Rhodococcus equi*

**DOI:** 10.3390/ani12101303

**Published:** 2022-05-19

**Authors:** Reinhard Sting, Ingo Schwabe, Melissa Kieferle, Maren Münch, Jörg Rau

**Affiliations:** 1Chemical and Veterinary Analysis Agency Stuttgart (CVUAS), Schaflandstr. 3/3, 70736 Fellbach, Germany; ingo.schwabe@cvuas.bwl.de (I.S.); melissa.kieferle@cvuas.bwl.de (M.K.); maren.muench@cvuas.bwl.de (M.M.); joerg.rau@cvuas.bwl.de (J.R.); 2Consiliary Laboratory for *Corynebacterium pseudotuberculosis* (DVG), Schaflandstr. 3/3, 70736 Fellbach, Germany

**Keywords:** *Rhodococcus equi*, alpaca, pyogranulomatous pneumonia, *vapA* gene, DNA sequencing, MALDI-TOF MS, antibiotic susceptibility

## Abstract

**Simple Summary:**

Serious consequences of septicemic bacterial infections include the formation of purulent and pyogranulomatous inflammation resulting in abscesses in inner organs. Different bacteria are known to cause these infections in livestock. In this study, we report in detail on a case of a fatal *Rhodococcus* (*R.*) *equi* infection in an alpaca (*Vicugna pacos*), to our knowledge, for the first time. *R. equi* is a member of the actinomycetes, a bacterial group known to contain several pathogenic bacteria. *R. equi* primarily affects equine foals and other domestic animals, but also humans, which renders this bacterium a zoonotic agent. The rhodococcal infection of the alpaca reported herein caused septicemia, resulting in emaciation and severe lesions in the lungs and heart. The onset of infection was presumably caused by aspiration pneumonia, resulting in abscesses exclusively in the lungs. The *R. equi* isolate proved to be pathogenic, based on the virulence gene *vapA* encoding the virulence-associated protein A. Antibiotic susceptibility testing revealed a susceptibility to doxycycline, erythromycin, gentamycin, neomycin, rifampicin, trimethoprim/sulfamethoxazole, tetracycline and vancomycin. This report of an *R. equi* infection in an alpaca makes clear that we still have knowledge gaps about bacterial infectious diseases in alpacas and potential zoonotic impacts. Therefore, the determination of pathogenic, zoonotic bacteria in alpacas is essential for treatment and preventive measures with respect to sustaining the health, welfare and productivity of this camelid species.

**Abstract:**

*Rhodococcus* (*R.*) *equi* is a pathogen primarily known for infections in equine foals, but is also present in numerous livestock species including New World camelids. Moreover, *R. equi* is considered an emerging zoonotic pathogen. In this report, we describe in detail a fatal rhodococcal infection in an alpaca (*Vicugna pacos*), to our best knowledge, for the first time. The alpaca died due to a septicemic course of an *R. equi* infection resulting in emaciation and severe lesions including pyogranulomas in the lungs and pericardial effusion. The onset of the infection was presumably caused by aspiration pneumonia. *R. equi* could be isolated from the pyogranulomas in the lung and unequivocally identified by MALDI-TOF MS analysis and partial sequencing of the 16S rRNA gene, the 16S-23S internal transcribed spacer (ITS) region and the *rpoB* gene. The isolate proved to possess the *vapA* gene in accordance with tested isolates originating from the lungs of infected horses. The *R. equi* isolates revealed low minimal inhibitory concentrations (MIC values) for doxycycline, erythromycin, gentamycin, neomycin, rifampicin, trimethoprim/sulfamethoxazole, tetracycline and vancomycin in antibiotic susceptibility testing. Investigations on the cause of bacterial, especially fatal, septicemic infections in alpacas are essential for adequately addressing the requirements for health and welfare issues of this New World camelid species. Furthermore, the zoonotic potential of *R. equi* has to be considered with regard to the One Health approach.

## 1. Introduction

*Rhodococcus* (*R.*) *equi* is a well-known pathogen primarily affecting foals. Rhodococcal infections cause serious pyogranulomatous bronchopneumonia, lymphadenitis, enteritis and lesions in internal organs as a consequence of septicemia [1,2]. However, besides horses, several other livestock and companion animals such as pigs, cattle, goats, cats and dogs are susceptible to *R. equi*, leading to a purulent infection especially in the lungs and lymph nodes, but also in other inner organs [3,4,5,6,7,8,9]. Case reports on *R. equi* infections in camelids have been presented for dromedaries [5] and llamas [10,11], but not yet, to our best knowledge, for alpacas.

Increasing numbers of alpacas kept in Europe for hobby, animal-aided therapy and commercial reasons require adequate diagnostic, therapeutic and preventive regimes to address health and welfare issues. Infections causing abscesses or pyogranulomas in inner organs are serious and often entail a debilitating and fatal end. *Corynebacterium* (*C.*) *pseudotuberculosis* is the most important pathogen associated with multiple external and internal abscesses in New World camelids [12,13,14]. However, several other pathogenic aerobic and anaerobic bacteria have been encountered in purulent infections and abscesses of New World camelids. Reports cover a broad spectrum of bacteria that include *Actinomyces* sp. [15,16], *Bibersteinia trehalosi*, *Schaalia* sp., *Trueperella pyogens* [16], *Mannheimia haemolyica* [17], *Actinobacillus* sp. [18], *Streptococcus agalactiae* [19], the *Streptococcus equi* subspecies *zooepidemicus* [20,21] and *Mycobacterium bovis* [22,23]. Therefore, investigations considering a broad spectrum of bacteria causing purulent infections and abscesses are essential for therapeutic options, and the prognosis and assessment of health risks.

*R. equi* is classified as a member of the Nocardiaceae family in the order of Actinomycetales. Due to confusion regarding the nomenclature of *Corynebacterium hoagie*, *R. equi*, and *R. hoagii,* we follow the recommendation of Vázquez-Boland et al. (2020) [24] by using the name *R. equi*.

Pathogenic *R. equi* can be found in pathogen-shedding animals and in soil considered to be a source of infection [25]. Hence, animals acquire rhodococcal infections by animal-to-animal transmission or from the soil, primarily via inhalation or ingestion [25,26,27,28]. In pathogenic *R. equi*, three virulence plasmid types have been described. The circular plasmids of the type pVAPA and pVAPB are associated with equine and porcine isolates, while the circular plasmid of the type pVAPN is associated with ruminants [29]. These plasmids carry the virulence genes *vapA*, *vapB* and *vapN*, respectively. These virulence-associated proteins (Vap) are essential for intramacrophage replication and disease development [30]. The *vapA* gene could be detected in *R. equi* isolates responsible for clinical diseases in equines and camelids [5,11,31]. However, human *R. equi* isolates carry each of these host-associated plasmid types [32], which suggests that animals are a source of infection in humans, rendering *R. equi* a zoonotic pathogen [33]. Therefore, the assessment of the pathogenicity of equine and camelid *R. equi* isolates is possible by testing the virulence-associated protein gene *vapA*. All strains of *R. equi* isolated from diseased horses [31] and camelids [5,11] possess a large plasmid carrying the *vapA* gene.

Furthermore, it has to be noted that *R. equi* is considered an emerging zoonotic pathogen. Rhodococcal infections mainly affect immunocompromised humans, due to HIV-infection, chemotherapy or organ transplantation, mainly affecting the lungs [34,35,36,37]. The zoonotic potential of this pathogen should particularly be noted in connection with animal associated infections, as a consequence of occupational or recreational exposure to farming and livestock or dry soil fertilized with manure from herbivores [29,38].

## 2. Materials and Methods

The 11-year-old male alpaca in question was born on 24 November 2010 and lived on a farm in Germany, in a group of 23 stallions that were part of a herd comprising about 77 animals. The alpaca stallion fell ill for the first time on 10 March 2021 with nasal discharge and purulent sinusitis and was treated with tetracycline. This alpaca was the only one to fall ill with those symptoms. About two weeks later, the stallion had a fever (39.8 °C) and was treated another time with enrofloxacin and meloxicam on 25 and 29 March 2021. Due to the deterioration of the animal’s condition, a treatment with amoxicillin, tulathromycin, meloxicam and dexamethasone and an inhalation therapy with a saline solution followed in a veterinary hospital from 30 March to 11 April 2021. Since the animal did not recover, it was released from the veterinary hospital. Later, the alpaca suffered from neurological symptoms resulting in dysphagia, cachexia and emaciation and eventually died on 16 May 2021. The carcass was submitted for pathological-anatomical and histopathological examinations on the same day. For the histopathological analysis, tissue samples were stained with hematoxylin and eosin.

Routine bacteriological examinations including incubation under aerobic and anaerobic conditions were carried out on organ specimens from the lung, liver, spleen, kidney, mesenteric lymph nodes and heart. The aseptically cut surfaces of the organ specimens were streaked directly onto Columbia agar supplemented with 5% sheep blood and onto MacConkey agar (BD, Heidelberg, Germany) and water-blue metachrome-yellow lactose agar according to Gassner (Oxoid, Wesel, Germany). In addition, specimens of the lungs were smeared on Pasteurella selective agar (Oxoid, Wesel, Germany). In addition, contents of the lung abscesses were streaked on CNA agar, a selective agar for Gram-positive bacteria (Columbia sheep blood agar supplemented with colistin and nalidixic acid; BD Heidelberg, Germany). The agar plates were incubated at 37 °C under aerobic and anaerobic (lung abscesses) conditions for at least 48 h. For cultivation of anaerobic bacteria, Schaedler agar and Wilkins–Chalgren agar with amikacin (BD, Heidelberg, Germany) were inoculated. Bacteria isolated in pure cultures were analyzed by MALDI-TOF mass-spectra generated by the microflex LT System (Bruker Daltonik, Bremen, Germany) using the Bruker Biotyper software Version 3.1. Spectral data were evaluated with the Bruker Taxonomy database (DB 9.604 entries; Bruker Daltonik) and an in-house database extension (Status: 1010 entries) as previously described [39]. The in-house extension includes additional MALDI-TOF mass-spectra for the *R. equi* reference strain ATCC 33701 (horse lung) and four *R. equi* German field isolates, CVUAS 2998 and CVUAS 952 (each from horse lungs), CVUAS 4057 (pig placenta), CVUAS 5384.2 (cattle lymph nodes) and three further *R. equi* field isolates from horses, isolated in Japan [40]. MALDI TOF mass-spectra of these *R. equi*, including the spectra isolated from the diseased alpaca, are available on our information exchange site via the MALDI-TOF user platform (https://www.maldi-up.ua-bw.de (accessed on 14 March 2022) [39]. Cluster analysis was done by the Biotyper OC software (Bruker) with setting correlation for distance measure to build a score-oriented dendrogram in average linkage mode.

The spectroscopic identification of the pathogen was verified by molecular identification. For this, DNA was extracted from bacterial cell suspensions by heat and the cell free supernatant used for PCR. PCR products for partial sequencing of the 16S rRNA gene were generated by using the broad-spectrum primers 27-F (AGAGTTTGATCMTGGCTCAG) [41] and 1492-R_modified (TASGGHTACCTTGTTACGACTT), which had been designed on the basis of sequences provided by Fredriksson et al. (2013) [42]. The DNA sequence of the 16S-23S rRNA internal transcribed spacer (ITS) region was obtained by decoding the PCR product obtained by using the 16S primer 1492-F_modified (reverse complement sequence of the primer 1492-R_modified) and the 23S primer 189-R_modified (GGSTACTDAGATGTTTCASTTC) designed on the basis of data published by Hunt et al. (2006) [43]. PCR products for sequencing the *rpoB* gene were generated by using the primers C2700F and C3130R [44]. DNA sequencing was performed on demand by Microsynth (Balgach, Switzerland) using the PCR primers. The *R. equi* isolates were identified by comparison of the DNA sequences with sequence entries deposited in the GenBank (National Center for Biotechnology Information [NCBI] [45]) and the EzBioCloud databases [46].

Furthermore, the alpaca and German *R. equi* isolates were tested for urease and nitrate reductase activity, synergistic hemolytic activity with *Staphylococcus aureus* (DSM 1104), *Listeria monocytogenes* (ATCC 19115), *Corynebacterium pseudotuberculosis* (field isolate CVUAS 10232) [47] and *Listeria ivanovii* (field isolate CVUAS 3382) [48]. The *R. equi* reference strain ATCC 33701 served as a positive control.

Detection of the virulence gene *vapA* was performed using the real-time PCR described by Harrington et al. (2005) [49], employing the *R. equi* reference strain ATCC 33701 as a positive control.

Minimum inhibitory concentrations (MIC) of antimicrobial drugs for the *R. equi* isolates were determined by the microdilution method, using the antibiotic microdilution plates MICRONAUT S Large animal and MICRONAUT-S Lifestock/Equines (GP) (MERLIN a Bruker Company, Bornheim-Hersel, Germany), which includes a layout designed for livestock and Gram-positive bacteria. Antimicrobial susceptibility testing was carried out according to the manufacturer’s instructions and recommendations of Riesenberg et al. (2013) [50] and evaluated using the MICRONAUT Software MCN 6 (MERLIN, Bornheim-Hersel, Germany).

## 3. Results and Discussion

### 3.1. Postmortem Examination

The carcass of the alpaca stallion had a low body weight of 49 kg compared to a normal average weight of about 80 kg. The pericardial and perirenal fat showed serious atrophy during the post mortem examination.

Further evident changes were found in the cranioventral part of the lungs, which showed a deep red to purple color and a solid consistency. Aspiration pneumonia and multifocal pea-sized pyogranulomas in the right lung became apparent. The tracheal mucosa showed a high degree of redness and the lymph nodes were very swollen. The pericardium contained a low-grade, clear, slightly reddish effusion. Moreover, strands of fibrin were noted in the thoracic cavity and pericardium.

Histopathological examinations of the lung showed high-grade, acute to subacute, multifocal to diffuse pyogranulomatous pneumonia. High numbers of alveolar macrophages, multifocal neutrophil granulocytes and individual plasma cells could be seen. Furthermore, numerous pyogranulomas with central cell detritus and numerous intralesional bacteria and partially enclosing plant material were visible (Figure 1).

The hepatocytes revealed dystrophy and a chronic proliferation of the bile tract with periportal fibrosis and acute hemorrhages. In those areas, the hepatocytes showed fibro-necrotizing changes. In the histological preparation of the gut, single cell necrosis in the epithelium of the villi and crypts and proliferation of the Peyer’s patches became evident.

Likewise, serious pathological-anatomical and histopathological changes and lesions have been found in the lungs of a llama [10] and in dromedaries [5] during post mortem examinations. However, Löhr et al. (2019) [11] reported on an *R. equi* infection in an 11-year-old llama severely affecting the small intestine (jejunum). The mesenteric lymph nodes, the liver and peritoneum were also involved. In all of the reported cases of *R. equi* infections in camelids, the macroscopic and microscopic (histological) changes were characterized by prominent pyogranulomatous to necrogranulomatous lesions and the formation of abscesses in the lungs and intestines. Lymph nodes, the pleura and peritoneum, and internal organs such as the liver and spleen were also affected [5,10,11]. There is evidence that survival is significantly reduced in equine foals infected with *R. equi* that develop extrapulmonary disorders in multiple organs, compared to foals that suffer from infections only affecting the lungs. However, it has to be considered that a clinical diagnosis of extrapulmonary manifestations is difficult and in many cases is not recognized until post mortem examination [51].

### 3.2. Bacteriological Examination

A bacteriological examination of material obtained from the lung pyogranulomas of the alpaca yielded a heavy growth of the *R. equi* isolate. In addition to the growth of *R. equi*, *Helcococcus ovis* and the obligate anaerobic bacteria *Fusobacterium necrophorum*, *Bacteroides fragilis* and *Prevotella heparinolytica* could be co-cultured. The concomitant growth of bacteria of the order Actinomycetales and the Gram-negative anaerobic bacteria *Fusobacterium necrophorum*, *Bacteroides* spp. and *Prevotella* spp. has also been previously encountered in bacteriological examinations of abscesses in camelids and ruminants [5,11,16,52].

*R. equi* could be identified by conventional methods. All *R. equi* isolates and the *R. equi* reference strain showed urease and nitrate reduction activity and synergistic hemolytic activities with *Staphylococcus aureus*, *Listeria monocytogenes*, *Corynebacterium pseudotuberculosis* and *Listeria ivanovii*. The identification based on classical methods was confirmed by MALDI-TOF MS, 16S rRNA gene, 16S-23S ITS and *rpoB* gene sequencing. In the MALDI-TOF MS analysis, a slightly higher score value was achieved for the *R. equi* isolate CVUAS 2755.3 from the alpaca due to the use of the supplemented database entries, in comparison to the commercial database (2.399 vs. 2.295). The isolates are reported as *R. hoagii* using the Bruker database. The MALDI-TOF MS dendrogram created by cluster analysis of spectra obtained by MALDI-TOF mass spectrometry shows a clear separation of the *Rhodococcus* species and a clear assignment of the isolate CVUAS 2755.3 to *R. equi* (Figure 2).

Partial sequencing of the 16S rRNA gene resulted in *R. equi* with a percentage identity difference of 0.6 (NCBI GenBank) or 0.7 (EzBioCloud database) to *R. soli*. However, 16S-23S ITS and rpoB sequencing revealed percent identity differences of 12.2% and 7.7%, respectively. Homology values of 100% and a minimum distance of differences in nucleotide sequences (SNPs) to the next best matching species of a 0.8% identity threshold allow the allocation of the bacterial species *R. equi* [53]. The partial 16S rRNA gene, 16S-23S ITS, and rpoB gene DNA sequences have been deposited into the NCBI GenBank with the accession numbers OM996150, ON003979 and ON009447, respectively.

The *vapA* gene, a characteristic feature of virulent *R. equi* strains, could be detected by PCR in accordance with previous findings for all clinical diseases in horses [24] and camelids [5,11]. While detection of the *vapA* gene in our study was successful in isolates originating in the alpaca and equine foals, this was not the case for the porcine and bovine isolate. Porcine and bovine isolates usually carry the *vapB* and *vapN* genes [8], which we did not test for in this study.

Conspicuously, fatal *R. equi* infections in camelids have been reported in current and previous cases exclusively for adult animals aged 2 to 11 years [5,10,11]. In contrast, adult horses are considered largely resistant to rhodococcosis [54]. However, from time to time fatal respiratory and intestinal *R. equi* infections have also been reported for adult horses in different countries [55,56,57].

The infection route and source of the pathogen remain puzzling. However, the pronounced and uniform changes in the respiratory and digestive tract of adult horses [55,56,57] and adult camelids [5,10,11] indicate an infection route via inhalation and/or ingestion. Remarkably, in our and another case of a llama fatality [11], plant material was detected in pyogranulomas in the lungs and intestines, respectively. This observation suggests an environmental origin of the pathogen of this air- and soil-borne pathogen, absorbed by aspiration or ingestion [28].

A comprehensive study performed in Australia revealed various factors favoring the pathogen load and infection in foals [58]. This study, performed on foals in stud farms comes to the conclusion that the main infection route of *R. equi* is via the respiratory tract. The inhalation of contaminated dust or direct foal to foal transmission through aerosolized virulent *R. equi* in high concentrations proved to be the most relevant factors for rhodococcal infections. This is especially the case in the stable, which is the most dangerous environment for the foals. It has been shown that the R. equi concentration inside the stables is ten times higher than in dusty paddock pastures [59]. Nevertheless, environmental conditions such as dry areas with low pasture cover are concomitant to significantly higher airborne concentrations of virulent *R. equi* and, as a consequence, increased infections of the respiratory tract of foals. As a result, clinically and subclinically infected foals exhale high concentrations of virulent *R. equi* into the air and in turn infect other animals.

Nevertheless, the primary source of infection in our case also remains unclear, all the more so because no direct contact with horses had occurred. However, the hay meadows had been fertilized with horse manure and horses and cows had been kept in the stables until they began, and have continued, to be used for the alpaca keeping in 2013.

### 3.3. Antibiotic Susceptibility

Antibiotic susceptibility testing of the *R. equi* isolates in vitro using the microdilution method revealed susceptibility to doxycycline, erythromycin, gentamycin, neomycin, rifampicin, trimethoprim/sulfamethoxazole, tetracycline and vancomycin. However, resistance or intermediate susceptibility was found to the penicillines ampicillin, oxacillin, and penicillin G, the cephalosporines cefquinome, ceftiofur, cefazolin, and cefoxitin, clindamycin, enrofloxacin, florfenicol and nitrofurantoin (Table 1). MIC values (MIC90) in a range comparable to our results have been previously reported by Riesenberg et al. (2014) [60]. In our study, we could not detect significant differences between the MIC values of the alpaca, equine, porcine or bovine *R. equi* strains. However, resistance to the antibiotics used for standard therapy and the emergence of multi-drug resistant *R. equi* isolates have been recently noted, thus limiting the available spectrum of effective antibiotics [61,62]. Due to infections caused by isolates resistant to macrolides and rifampicin, an alternative use of antibiotics for the treatment of rhodococcal infections has been critically reviewed by Cisek et al. (2014) [63]. The authors recommend treating rhodococcosis with a combination of at least two antibiotics including macrolides, rifampicin, fluoroquinolones, aminoglycosides, glycopeptides and carbapenems. Furthermore, the macrolides can cause life-threatening hyperthermia caused by drug-induced anhidrosis and diarrhea in equine foals and fatal colitis in mares [64].

In this respect, guidelines for the prudent use of antibiotics in veterinary medicine should be taken into account to avoid ineffective antimicrobial treatment. A recent study on the treatment of foals suffering from pneumonia caused by *R. equi* concluded that foals with smaller lesions could even be spared from antibiotic treatment without a significant increase in mortality [65]. Whether and to what extent this treatment regime also applies to alpacas still needs to be investigated.

In order to prevent *R. equi* infections and subsequent antibiotic treatment, effective prophylactic and preventive management practices have been discussed. However, no universal prevention strategy has so far been proven effective enough against rhodococcal infections [1,66].

## 4. Conclusions

This is the first detailed report of a fatal *R. equi* infection in an adult alpaca, to our best knowledge. The present report extends our knowledge about *R. equi* infection in alpacas, the most important New World camelid species kept under human care. Thus, this report should be of particular concern for the health and husbandry of this New World camelid species living in Europe. Furthermore, this bacterium is a zoonotic agent. The source of infections of this soil-and-airborne pathogen is suspected to be environmental and its transmission caused by aspiration or ingestion. However, transmission via close contact between animals and from animal to humans might also be of relevance. Therefore, reports on *R. equi* infections are important for assessing the relevance of pathogen reservoirs and transmission routes as a basis of effective measures in order to sustain the health, productivity and welfare of alpacas.

## Figures and Tables

**Figure 1 animals-12-01303-f001:**
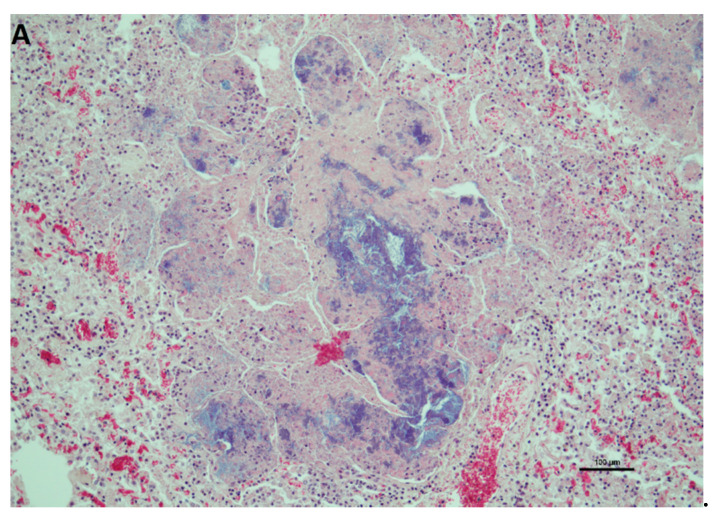
Histological section of a focal pyogranuloma in the lung of the alpaca with central cell debris as well as peripheral neutrophil granulocytes and alveolar macrophages (**A**), and plant material (**B**). Numerous bacteria, both extra- and intracellular in neutrophils and macrophages are also pictured (magnification 100×).

**Figure 2 animals-12-01303-f002:**
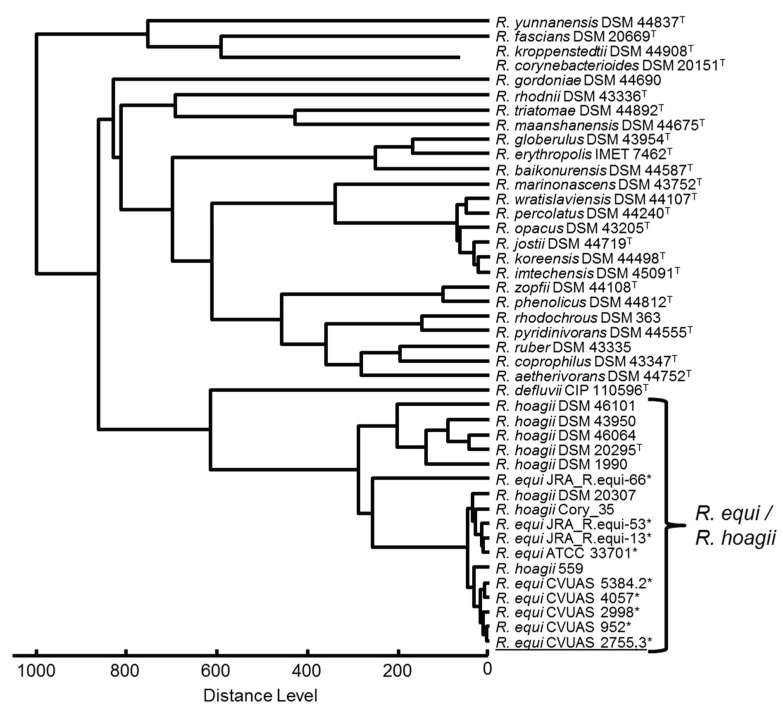
MSP Dendrogram created by cluster analysis of reference main spectra (MSP) obtained by MALDI-TOF mass spectrometry (MALDI Biotyper, Version 3.1, Bruker Daltonik) of a *Rhodococcus* (*R.*) *equi* isolate from an alpaca (underlined) in comparison to a selection of available MSPs of strains of the genus *Rhodococcus* from the commercial Bruker Biotyper database and external reference entries, documented in the MALDI-UP catalogue on https://maldi-up.ua-bw.de (accessed on 14 March 2022) [39]. Type strains (T) and external reference spectra (*) are indicated. In context of this study, the following self-created MSPs were used for comparison: *R. equi* reference strain ATCC 33701 (horse lung) and the four *R. equi* German field isolates, CVUAS 2998 and CVUAS 952 (each from horse lungs), CVUAS 4057 (pig placenta), and CVUAS 5384.2 (cattle lymph nodes).

**Table 1 animals-12-01303-t001:** Determination of minimal inhibitory concentration (MIC) values (mg/L) for *R. equi* isolates using the micro-dilution method. Assessment of the MIC values was carried out using the MCN6 software (Merlin, Germany). R = resistance, I = intermediary susceptibility, S = susceptibility. n.d.p. = no data provided.

Antibiotic	CVUAS 2755.3 (Alpaca, Lung)	CVUAS 246 (ATCC 33701)	CVUAS 2998 (Horse, Lung)	CVUAS 952 (Horse, Lung)	CVUAS 4057 (Pig, Placenta)	CVUAS 5384-2 (Cattle, Lymph Node)	MIC_50_ ^1^	MIC_90_ ^1^
Ampicillin	=8 (R)	=8 (R)	=8 (R)	=8 (R)	=8 (R)	=8 (R)	4	8
Cefquinom	>4 (R)	>4 (R)	=4 (I)	=4 (I)	=4 (I)	>4 (R)	2	4
Ceftiofur	>4 (R)	>4 (R)	>4 (R)	>4 (R)	>4 (R)	>4 (R)	8	16
Cefazolin	>8 (R)	>8 (R)	>8 (R)	>8 (R)	>8 (R)	>8 (R)	n.d.p.	n.d.p.
Clindamycin	>2 (R)	>2 (R)	>2 (R)	>2 (R)	>2 (R)	>2 (R)	4	8
Cefoxitin	>4 (R)	=4 (S)	>4 (R)	>4 (R)	>4 (R)	>4 (R)	n.d.p.	n.d.p.
Doxycyclin	=0.5 (S)	=0.5 (S)	=0.5 (S)	=0.5 (S)	=0.5 (S)	=0.5 (S)	1	1
Enrofloxacin	=1 (I)	=0.5 (I)	=0.5 (I)	=0.5 (I)	=0.5 (I)	=0.5 (I)	1	1
Erythromycin	=0.5 (S)	=0.5 (S)	=0.5 (S)	=0.5 (S)	=0.5 (S)	=0.5 (S)	0.5	0.5
Florfenicol	=8 (R)	>8 (R)	>8 (R)	>8 (R)	=8 (R)	>8 (R)	16	16
Gentamicin	≤1 (S)	≤1 (S)	≤1 (S)	≤1 (S)	≤1 (S)	≤1 (S)	0.5	0.5
Neomycin	≤8 (S)	≤8 (S)	≤8 (S)	≤8 (S)	≤8 (S)	≤8 (S)	n.d.p.	n.d.p..
Nitrofurantoin	=64 (I)	=64 (I)	=64 (I)	=64 (I)	=64 (I)	=64 (I)	n.d.p.	n.d.p..
Oxacillin	>4 (R)	>4 (R)	>4 (R)	>4 (R)	>4 (R)	>4 (R)	n.d.p.	n.d.p.
Penicillin G	>8 (R)	>8 (R)	>8 (R)	=8 (R)	=8 (R)	=8 (R)	4	4
Rifampicin	=0.5 (S)	=0.5 (S)	=0.5 (S)	=0.25 (S)	=0.5 (S)	=0.25 (S)	0.06	0.12
Trimethoprim/Sulfamethoxazole ^2^	=1 (S)	=1 (S)	=1 (S)	=2 (S)	=1 (S)	=1 (S)	0.25	0.5
Tetracycline	=2 (S)	=4 (S)	=2 (S)	=2 (S)	=2 (S)	=2 (S)	4	8
Vancomycin	≤0.5 (S)	≤0.5 (S)	≤0.5 (S)	≤0.5 (S)	≤0.5 (S)	≤0.5 (S)	0.5	0.5

^1^ MIC50 and MIC90 values adapted from Riesenberg et al. (2014) [60]. ^2^ The MIC values of trimethoprim/sulfamethoxazole (1:19) are expressed as the MIC values of trimethoprim.

## Data Availability

Partial 16S rRNA gene, 16S-23S ITS and rpoB gene DNA sequences have been deposited into the NCBI GenBank with the accession numbers OM996150, ON003979 and ON009447, respectively. MALDI-TOF MS reference entries of *R. equi* isolates generated in the present study, as well as single spectra for validation purposes, are available for exchange via the MALDI-UP homepage (https://www.maldi-up.ua-bw.de (accessed on 14 March 2022) [39].

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
