# Peer review of "Fatal Infection in an Alpaca (Vicugna pacos) Caused by Pathogenic Rhodococcus equi"

_animals, 2022, doi:10.3390/ani12101303_

Round 1
Reviewer 1 Report
The study proposes the bacterium Rhodococcus equi as a pathogenic agent for alpacas. This unambiguously an important finding which however is not completely supported. The authors should add a figure not only from the tissues under microscope but also from the external examination from the animal and better describe the macroscopical signs on it. Thus I recommend major revision according to the aforementioned and the following recommendations
lines 53-54: “We are reporting here…”. This statement belongs to the results or discussion and should be moved there
In the introduction the authors refer to several cases of R. equi infections. Are all these cases real infections or occasionally just identification of the pathogen in those animals?
lines 55-58: In which countries, everywhere? Please be more specific
Is there a human pathogenic potential of R. equi? Any related infections?
lines 80-81: Again this is a finding and should be transferred in the discussion
In Materials and methods there s no information regarding DNA isolation. Please supplement
lines 138-140: please describe more details regarding the positive control. How was it provided?
In line 151 please report the normal weight of an alpaca. Not all readers are familiar
In the results, additionally, some images of the MALDI-TOF profiles would be indicative and should be added
Author Response
Assigned Editor: Cherry Huang
Journal: Animals
Manuscript Status: Pending major revisions
Manuscript ID: animals-1666789
Type: Case Report
Title: Fatal Infection in an Alpaca (Vicugna pacos) Caused by Pathogenic Rhodococcus equi
Authors
Reinhard Sting, Ingo Schwabe, Melissa Kieferle, Maren Münch, Jörg Rau
Section: Animal Welfare
Special Issue: Trends in Camel Health and Production
Reviewer 1
The study proposes the bacterium Rhodococcus equi as a pathogenic agent for alpacas. This unambiguously an important finding which however is not completely supported. The authors should add a figure not only from the tissues under microscope but also from the external examination from the animal and better describe the macroscopical signs on it. Thus I recommend major revision according to the aforementioned and the following recommendations
Authors‘ Response:
Unfortunately, no pictures were taken during the post mortem examination.
lines 53-54: “We are reporting here…”. This statement belongs to the results or discussion and should be moved there
Authors‘ Response:
This sentence has been moved from the Chapter „Introduction“ to the chapter „Conclusion“.
In the introduction the authors refer to several cases of R. equi infections. Are all these cases real infections or occasionally just identification of the pathogen in those animals?
Authors‘ Response:
The cases of R. equi infections reported for domestic animals are associated with purulent infection especially in the lungs and lymph nodes but also in other inner organs. This information has been added in the chapter “Introduction”.
lines 55-58: In which countries, everywhere? Please be more specific
Authors‘ Response:
„in Europe“ has been supplemented.
Is there a human pathogenic potential of R. equi? Any related infections?
Authors‘ Response:
- equi infection have a strong correlation to HIV infections. This aspect has been considered: Rhodococcal infections mainly affect immunocompromised humans, due to HIV-infection, chemotherapy or organ transplantation, mainly affecting the lungs [34,35,36,37].
lines 80-81: Again this is a finding and should be transferred in the discussion
Authors‘ Response:
The statement in lines 80-81 has been moved to the chapter „Conclusion“.
In Materials and methods there s no information regarding DNA isolation. Please supplement
Authors‘ Response:
This point of criticism is in contrast to reviewer 1. Therefore, only basic details on DNA extraction have been provided.
Lines 138-140: please describe more details regarding the positive control. How was it provided?
Authors‘ Response:
The R. equi strain ATCC 33701 was purchased from ATCC. The use of the reference strain ATCC 33701 has been specified in the chapter “Material and Methods” and “Results”.
In line 151 please report the normal weight of an alpaca. Not all readers are familiar
Authors‘ Response:
The a normal average weight of an alpaka stallion is about 80 kg. This piece of information has been supplemented in the text.
In the results, additionally, some images of the MALDI-TOF profiles would be indicative and should be added
Authors‘ Response:
A MALDI MSP dendrogram created by cluster analysis of spectra obtained by MALDI-TOF mass spectrometry has been included as Figure 2.

Reviewer 2 Report
The article describes an interesting case. However, it should be rewritten.
In this form, the text contains too many repetitions, for example, regarding the virulence of R. equi (lines 71-72, 72, 187, 211-214). However, despite the replays, virulence factors and plasmid type as not described well enough. Therefore, it should be shortened, placed in the introduction, and mentioned in the discussion.
The PCR diagnostic, MALDI-TOF, and MIC analysis are described in detail and largely present in the text. However, that information does not provide any essential new data. Therefore, this part should be shortened and described in a more compressive form.
On the other side, the article shows a clinical case. The history of the animal should be better described. For how long has this animal suffered weakness? Any data regarding previous diagnoses or treatments? Was this individual born in Germany or imported? What about the health status of other animals in the herd? Last but not least, the necropsy photos are lacking.
The literature review regarding purulent lesions in alpacas or other camelids should be added to the text. R. equi infection is rare in animals other than foals. Purulent lesions in farm and wild animals are associated with infections with pathogens such as Corynebacterium spp., mostly pseudotuberculosis, and Trueperella (Arcanobacterium) pyogenes; Staphylococcus spp., Streptococcus spp., Pseudomonas aeruginosa, Moraxella spp. and of course Rhodococcus equi. Moreover, lesions caused by those pathogens often appear indistinguishable from typical Mycobacterium-associated lesions.
For example, R. equi was isolated from cattle and American bison with purulent lesions suspected of Mycobacterium spp. Infection. In pigs, R. equi, not previously suspected Mycobacterium spp., is the primary causal agent of lymphadenitis. Furthermore, it was suggested that Mycobacterium spp. infection could predispose farm and wild animals to R. equi infection.
Krajewska-Wędzina M, Didkowska A, Sridhara AA, Elahi R, Johnathan-Lee A, Radulski Ł, Lipiec M, Anusz K, Lyashchenko KP, Miller MA, Waters WR. Transboundary tuberculosis: Importation of alpacas infected with Mycobacterium bovis from the United Kingdom to Poland and potential for serodiagnostic assays in detecting tuberculin skin test false-negative animals. Transbound Emerg Dis. 2020 May;67(3):1306-1314. doi: 10.1111/tbed.13471. Epub 2020 Feb 4. PMID: 31899584.
Other suggested corrections:
Line 47 – equine foals, please correct to foals
Line 151-153 – This description should be shortened and rewritten. The adult male alpaca has bodyweight et least 70-80 kg. 50 kg carcass is not suggesting the emaciation. It is a precise diagnosis and doesn't need to be confirmed by measuring the fat tissue.
Line 155 –An aspiration pneumonia is a speculation. The infection might be spread with blood also.
Line 232-236 – the stable is the most dangerous environment for the foals, is has been shown that R. equi concentration inside the stables Is ten times higher than in dusty paddock pastures.
Bordin AI, Huber L, Sanz MG, Cohen ND. Rhodococcus equi foal pneumonia: Update on epidemiology, immunity, treatment and prevention. Equine Vet J. 2022 May;54(3):481-494. doi: 10.1111/evj.13567. Epub 2022 Mar 21. PMID: 35188690.
Line 252 – for et least two decades, erythromycin is not recommended for rhodococcosis treatment because of severe adverse effects, primarily lethal diarrhea
Line 272-274 vaccine is still unavailable; the continuation of the studies described in cited articles (59, 60) showed that the PNAG vaccine candidate failed in the further research, see:
Kahn SK, Cywes-Bentley C, Blodgett GP, Canaday NM, Turner-Garcia CE, Flores-Ahlschwede P, Metcalfe LL, Nevill M, Vinacur M, Sutter PJ, Meyer SC, Bordin AI, Pier GB, Cohen ND. Randomized, controlled trial comparing Rhodococcus equi and poly-N-acetyl glucosamine hyperimmune plasma to prevent R equi pneumonia in foals. J Vet Intern Med. 2021 Nov;35(6):2912-2919. doi: 10.1111/jvim.16294. Epub 2021 Nov 5. PMID: 34738651; PMCID: PMC8692225.
Author Response
Assigned Editor: Cherry Huang
Journal: Animals
Manuscript Status: Pending major revisions
Manuscript ID: animals-1666789
Type: Case Report
Title: Fatal Infection in an Alpaca (Vicugna pacos) Caused by Pathogenic Rhodococcus equi
Authors
Reinhard Sting, Ingo Schwabe, Melissa Kieferle, Maren Münch, Jörg Rau
Section: Animal Welfare
Special Issue: Trends in Camel Health and Production
Reviewer 2
The article describes an interesting case. However, it should be rewritten.
In this form, the text contains too many repetitions, for example, regarding the virulence of R. equi (lines 71-72, 72, 187, 211-214). However, despite the replays, virulence factors and plasmid type are not described well enough. Therefore, it should be shortened, placed in the introduction, and mentioned in the discussion.
Authors‘ Response:
The text “of a virulent, the vapA gene bearing” has been removed in the sub-chapter 3.2 Bacteriological examination. The paragraph from line 208 to 219 has been shortened.
The virulence plasmid types and the the vap genes have been described in the chapter “Introduction” and mentioned in the chapter “Result and Discussion”.
The PCR diagnostic, MALDI-TOF, and MIC analysis are described in detail and largely present in the text. However, that information does not provide any essential new data. Therefore, this part should be shortened and described in a more compressive form.
Authors‘ Response:
This point of criticism is in contrast to Reviewer 1. There for only basic details on the PCR procedure are provided in the chapter “Material and Methods”.
The paragraph on “Antibiotic susceptibility“ has been shortened.
On the other side, the article shows a clinical case. The history of the animal should be better described. For how long has this animal suffered weakness? Any data regarding previous diagnoses or treatments? Was this individual born in Germany or imported? What about the health status of other animals in the herd? Last but not least, the necropsy photos are lacking.
Authors‘ Response:
More detailed information on the history of the animal has been added in the chapter “Materials and Methods”.
Unfortunately, no pictures were taken during the post mortem examination.
The literature review regarding purulent lesions in alpacas or other camelids should be added to the text. R. equi infection is rare in animals other than foals. Purulent lesions in farm and wild animals are associated with infections with pathogens such as Corynebacterium spp., mostly pseudotuberculosis, and Trueperella (Arcanobacterium) pyogenes; Staphylococcus spp., Streptococcus spp., Pseudomonas aeruginosa, Moraxella spp. and of course Rhodococcus equi. Moreover, lesions caused by those pathogens often appear indistinguishable from typical Mycobacterium-associated lesions.
For example, R. equi was isolated from cattle and American bison with purulent lesions suspected of Mycobacterium spp. Infection. In pigs, R. equi, not previously suspected Mycobacterium spp., is the primary causal agent of lymphadenitis. Furthermore, it was suggested that Mycobacterium spp. infection could predispose farm and wild animals to R. equi infection.
Krajewska-Wędzina M, Didkowska A, Sridhara AA, Elahi R, Johnathan-Lee A, Radulski Ł, Lipiec M, Anusz K, Lyashchenko KP, Miller MA, Waters WR. Transboundary tuberculosis: Importation of alpacas infected with Mycobacterium bovis from the United Kingdom to Poland and potential for serodiagnostic assays in detecting tuberculin skin test false-negative animals. Transbound Emerg Dis. 2020 May;67(3):1306-1314. doi: 10.1111/tbed.13471. Epub 2020 Feb 4. PMID: 31899584.
Authors‘ Response:
The aspect of further bacteria causing abscesses has been considered in more depth in the chapter “Introduction”.
Other suggested corrections:
Line 47 – equine foals, please correct to foals
Authors‘ Response:
“equine foals” has been replaced by “foals”.
Line 151-153 – This description should be shortened and rewritten. The adult male alpaca has bodyweight et least 70-80 kg. 50 kg carcass is not suggesting the emaciation. It is a precise diagnosis and doesn't need to be confirmed by measuring the fat tissue.
Authors‘ Response:
The first paragraph of the sub-chapter “3.1 Postmortem examination” has been rewritten.
Line 155 –An aspiration pneumonia is a speculation. The infection might be spread with blood also.
Authors‘ Response:
The statement has been weakened by the formulation „This observation suggests …“.
Line 232-236 – the stable is the most dangerous environment for the foals, is has been shown that R. equi concentration inside the stables Is ten times higher than in dusty paddock pastures.
Bordin AI, Huber L, Sanz MG, Cohen ND. Rhodococcus equi foal pneumonia: Update on epidemiology, immunity, treatment and prevention. Equine Vet J. 2022 May;54(3):481-494. doi: 10.1111/evj.13567. Epub 2022 Mar 21. PMID: 35188690.
Authors‘ Response:
The authors thank the reviewer for this important aspect. This aspect has been included in the text.
Line 252 – for at least two decades, erythromycin is not recommended for rhodococcosis treatment because of severe adverse effects, primarily lethal diarrhea.
Authors‘ Response:
We thank the reviewer for this important addition. We have included this information in the text.
Line 272-274 vaccine is still unavailable; the continuation of the studies described in cited articles (59, 60) showed that the PNAG vaccine candidate failed in the further research, see:
Kahn SK, Cywes-Bentley C, Blodgett GP, Canaday NM, Turner-Garcia CE, Flores-Ahlschwede P, Metcalfe LL, Nevill M, Vinacur M, Sutter PJ, Meyer SC, Bordin AI, Pier GB, Cohen ND. Randomized, controlled trial comparing Rhodococcus equi and poly-N-acetyl glucosamine hyperimmune plasma to prevent R equi pneumonia in foals. J Vet Intern Med. 2021 Nov;35(6):2912-2919. doi: 10.1111/jvim.16294. Epub 2021 Nov 5. PMID: 34738651; PMCID: PMC8692225.
Authors‘ Response:
This current research has been considered and the text adapted in the chapter „Discussion“.

Round 2
Reviewer 1 Report
Figure 2 does not the show MALDI-TOF profiles as I recommended, but instead a dendrogram that is not explained how exactly it was created. Regarding the remaining comments, I believe they were addressed.
Author Response
Creating spectra library (MSP) dendrograms of MALDI-TOF mass spectral profiles for identification of bacterial species and comparison of different species of the same genus is a recognized procedure. Single spectra are of only little value.The MSP dendrogram was created using a biotyper software.
We suggest to add the following sentence in the legend for Figure 1:
"Cluster analysis was done by the Biotyper OC software (Bruker) with setting correlation for distance measure to build a score-oriented dendrogram in average linkage mode."
I hope this is an adequate addition.